# Evolution of default genetic control mechanisms

**William Bains** [1,2]*, **Enrico Borriello**[3], **Dirk Schulze-Makuch**[4,5,6]

**1** Department of Earth, Atmospheric and Planetary Sciences, Massachusetts Institute of Technology, Cambridge, MA, United States of America, **2** School of Physics & Astronomy, Cardiff University, 4 The Parade, Cardiff, United Kingdom, **3** School of Complex Adaptive Systems, College of Global Futures, Arizona State University, Tempe, AZ, United States of America, **4** Zentrum für Astronomie und Astrophysik, Technische Universität Berlin, Berlin, Germany, **5** German Research Centre for Geosciences (GFZ), Section Geomicrobiology, Potsdam, Germany, **6** Department of Experimental Limnology, Leibniz-Institute of Freshwater Ecology and Inland Fisheries (IGB), Stechlin, Germany

* bains@mit.edu

**Data Availability Statement:** Model is fully available as run by the authors as a supplementary file to this submission.

**Funding:** The authors received no specific funding for this work.

## Abstract

We present a model of the evolution of control systems in a genome under environmental constraints. The model conceptually follows the Jacob and Monod model of gene control. Genes contain control elements which respond to the internal state of the cell as well as the environment to control expression of a coding region. Control and coding regions evolve to maximize a fitness function between expressed coding sequences and the environment. The model was run 118 times to an average of $1.4 \cdot 10^6$ 'generations' each with a range of starting parameters probed the conditions under which genomes evolved a 'default style' of control. Unexpectedly, the control logic that evolved was not significantly correlated to the complexity of the environment. Genetic logic was strongly correlated with genome complexity and with the fraction of genes active in the cell at any one time. More complex genomes correlated with the evolution of genetic controls in which genes were active ('default on'), and a low fraction of genes being expressed correlated with a genetic logic in which genes were biased to being inactive unless positively activated ('default off' logic). We discuss how this might relate to the evolution of the complex eukaryotic genome, which operates in a 'default off' mode.

## 1 Introduction

Obligate multicellularity is uniquely a eukaryotic trait [1–3], and with it the morphological complexity that comes from combining many distinct cell types into one organism. Multicellularity requires complex genetic controls both to provide the control to generate different genetic activity patterns in different cell types and to provide the 'programme' to construct the adult organism. In addition, the more complex internal architecture and controls in the eukaryotic cell also require specific controls. In some single-celled eukaryotes such internal complexity resembles that of equivalently sized multicellular organisms. Reflecting this, genome sizes in eukaryotes can exceed those of the largest bacterial or archaeal ("prokaryotic") genomes by 4 orders of magnitude (Fig 1).

**Competing interests:** The authors have declared that no competing interests exist.

There is substantial overlap in *coding capacity* between the larger prokaryotic genomes and eukaryotic genomes. The coding capacity of some of the larger prokaryotic genomes such as those of some cyanobacteria (~12,000 coding sequences (CDS) [4]), *Ktedonobacter racemifer* (~11,500 CDS [5]), *Sorangium cellulosum* (~9000 CDS [6]), *Magnetobacterium bavaricum* (~8500 CDS [7]) overlaps with coding capacity of multicellular fungi (5000–15,000 (e.g. [8,9]) and autotrophic protists (10,000–20,000 CDS [9]) and approaches that of *Drosophila melanogaster* (~16,000 CDS [10]). The size difference between prokaryotic and eukaryotic genomes is primarily due to non-coding DNA that is related in part to gene control. Thus the *E.coli* genome has little non-coding DNA, and ~285 proteins are involved in gene control [11], ~7% of the genome. By contrast over 90% of the human genome is non-coding, and conservative estimates are that 10 times as many non-coding bases as coding bases are evolutionarily conserved (i.e. are presumed to have selectable function unrelated to coding) [12,13]. Even *Saccharomyces cerevisiae* has 400 proteins associated with chromatin structure and function, as well as histones and polymerases [14], to control ~5300 genes [15] the majority of which have only core promoters and no regulatory elements [14], compared to E.coli's ~285 proteins to control 4300 genes [16].

What enabled this increase in genetic complexity? The key difference between prokaryotic and eukaryotic cells have been suggested to be chemistry, intracellular structure, energetics and genetics. In general, any small molecule structure made by a eukaryotic cell will be made by a prokaryote as well. Many 'eukaryotic' cellular structures are actually found in a few prokaryotes as well. Linear chromosomes are found in bacteria [17–20]. Intracellular membrane compartments for secretion and processing [21,22] and energy capture [22–27] as well as membrane-bound DNA-containing bodies are found in Planctomycetes [25,28].

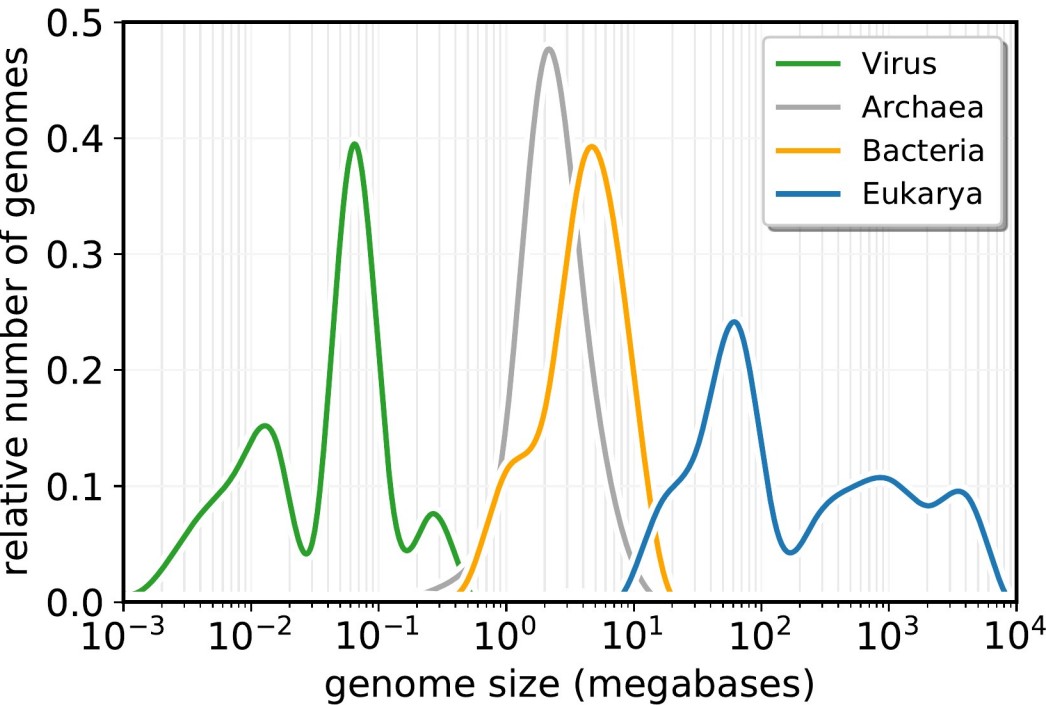

**Fig 1. Sizes of completed genome sequences, showing distinct size distributions for prokaryotes (bacteria and archaea), eukaryotes and viruses.** X axis: Genome size in megabases, Y axis: Fraction of each of the four classes of organism that have genome of that size. Data from https://www.ncbi.nlm.nih.gov/genome/browse/#!/overview/, accessed 15[th] June 2020; based on 27308 bacteria, 1769 Archaea, 5300 Eukarya and 19536 viruses.

*Achromatium oxaliferum* contains complex internal membranes containing calcium carbonate (whose function is obscure) [29], *Entotheonella* detoxifies arsenic and barium by sequestering it in internal vesicles [30], and cyanobacteria have stacked internal photosynthetic membranes [31]. The intracellular membrane system of eukaryotes is integrated into a dynamic network of vesicle trafficking and control which is rare in prokaryotes (reviewed in [32]); however some of the core proteins and structural elements of a cytoskeleton are also found in prokaryotes [33–38], and the giant bacterium *Epulopiscium fishelsoni* has an internal tubule system so similar to eukaryotes that it was initially mistaken for a protozoan [39,40]. These examples all suggest that complex structure *per se* follows from large size, rather than large size following from internal structure.

It is widely accepted that the modern eukaryotic cell evolved by a series of endosymbiotic events [41,42]. Recent insights gained from molecular biology show examples of endosymbiotic bacteria that live inside other bacteria [43–46], and bacteria that live inside modern mitochondria [47] (as well as a wealth of endosymbiotic bacteria in eukaryotic cells) which suggests that prokaryotic endosymbiotic events, while unusual, are not extremely rare. Lane and Martin [48–50] suggest that the endosymbiotic event, and consequent development of internal membrane-bound energy-generating organelles, enabled the ability to generate energy from intracellular membranes acquired through endosymbiosis is key, as more genes imply more proteins and proteins require energy to make. We find this theory lacking for three reasons. Firstly, the majority of genes in the larger eukaryotic genomes do not code protein–complexity comes from non-coding RNA genes and regulatory elements as discussed above. Secondly, most of the coding genes in any one cell are not transcribed; indeed the whole reason to maintain a complex genetic apparatus is so that different subsets of genes can be expressed at different times. Genomes containing more coding sequences do not make more proteins *at any one time*. Lastly, protein synthesis is only the major use of cellular energy in autotrophic bacteria grown under conditions of unlimited nutrition. Under more normal conditions of growth, protein synthesis rarely is observed to consume more than 20% of the cell's energy, and of course in non-growing cells (which is most cells in the biosphere most of the time) protein synthesis is only needed for maintenance and turnover, a minor part of the overall 'maintenance energy' [51–53]. (See S1 File for a more detailed analysis protein synthesis' energy requirements).

We have recently suggested that the default logic of gene control is a significant factor enabling eukaryogenesis [54]. Here 'default logic' means whether a stretch of DNA downstream of a polymerase binding site is likely to expressed unless it is repressed ("Default on"), or whether it is not expressed unless it is activated ("default off"). It is observed that it is easier for a gene to be expressed in a prokaryote than a eukaryote, as evidenced by the fate of pseudogenes, the construction of expression vectors, and the fate of differentiated gene expression in cell fusion (reviewed in [55]) as well as arguments from the mechanisms of gene control (See below). While there are exceptions, it is broadly true that the eukaryotic genome is by default 'off' and needs metabolic energy to turn 'on', whereas in a prokaryotic genome genes are by default 'on' unless turned 'off'.

This default control mode is reflected in the thermodynamics of gene control. In eukaryotes, complexes of proteins are required to remodel chromatin around promoters before genes can be transcribed, involving the ATPase molecular motors Snf2 and Sth1 [56,57], and subsequent ATP-dependent binding of transcription factors to chromatin [58] before RNA polymerase can bind to a promoter. The remodelling involves (inter alia) ATP-dependent removal of H2A/B dimers from nucleosomes [59–62] (Archaeal nucleosomes lack H2A/B dimers, and consist of homologues of H3/4 dimers only [63,64]). By contrast the molecular rearrangements that control initiation of bacterial transcription are powered by the binding energy of the

various proteins [11,65,66]. Archaea have similar transcription initiation logic to bacteria, despite having RNA polymerase complexes similar to those in eukaryotes [67,68]. RNA elongation in bacteria requires roughly 1.5 ATP per base added, again being controlled by protein binding factors [11]. In Eukaryotes ATP-dependent chromatin remodelling is required for RNA elongation, as well as energy-consuming histone acetylation and methylation chemistry [69].

Default logic relates to the internal logic of control, and does not specify whether the genes in a genome *are* active at any one time. In prokaryotic spores almost all of the genes are unexpressed, (See e.g. [70–72]), despite them having 'default on' genetic control, and around 80% of the coding genes are expressed in mammalian testis [73], despite having 'default off' genetic control. 'Default off' genetic control makes it more metabolically costly to express a gene than 'default on', but does not specify whether a gene is expressed; specific expression patterns are determined by the function of the gene in the cell or organism.

However it is plausible to suggest that this 'Default off' logic is more efficient if the majority of the genome is silent, as would be the case if the genome encoded many expression programmes only one of which is active at once. It would also allow the facile accumulation of silenced duplicate genes to act as the substrate for genome complexification, which itself is associated with rapid diversification and adaptation [74] (although see [75]). We propose that a 'default off' logic will favour the evolution of complex genomes which code for multiple expression patterns, a 'default on' logic will favour the evolution of compact, efficient genomes with relatively few distinct phenotypes.

This hypothesis should be testable by simulation and by experiment. As a first step in this we present a simplified model of gene control and evolution that can evolve either 'Default On' or 'Default Off' logic. In this paper we present the model, and initial results from its execution.

## 2 Methods

### 2.1 Modelling approach

We attempt to model the evolution of control logic of genes under selective pressure. As a balance between the need for computability on one hand and the need for biological 'realism' on the other, we chose the 'classical' operon as a model on which to build the model structure. A series of sequences upstream of the coding sequence can bind proteins which allow, promote or catalyse transcription (positive elements) or which can bind proteins that retard or prevent transcription (negative elements). A similar process applies to eukaryotic genes in that positive and negative regulatory elements influence the transcription of the gene, although in eukaryotes those regulatory elements may be distant from the gene. Whether those regulatory elements are active will depend on the proteins in the cell, so that there is feedback between the phenotype and the transcription of the genotype that it encodes. The fitness of an organism depends on the 'fit' between its phenotype and its environment, but that environment can change, so the expression of genes must also be influenced by the environment. The model must also be able to be queried for some surrogate of 'default off' or 'default on' genetics *independent* of how many genes in an organism are actually transcribed at any one time (which will depend on the demands of the environment).

The properties of the model are summarised in Fig 2A.

### 2.2 Specifics of the model

To capture the requirements above, the model was constructed as follows. For simplicity, everything in the model is strings of one type. Thus the *phenotype* is a set of strings of the same

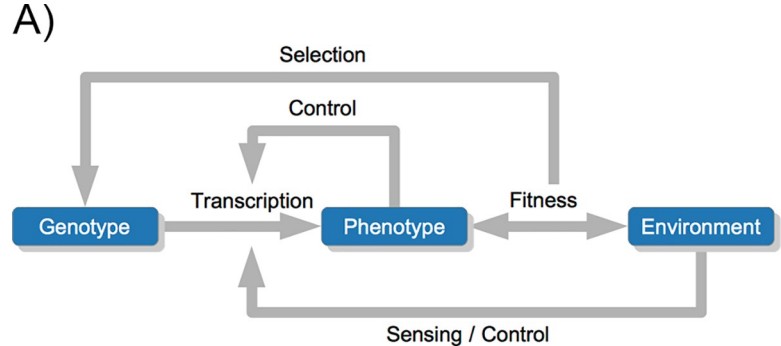

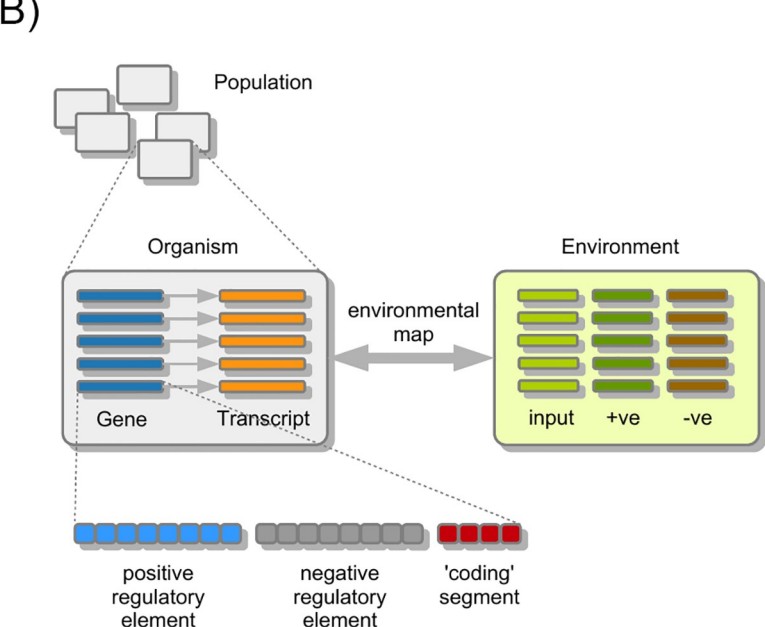

**Fig 2. Summary of model structure.** A: Overall design philosophy, showing feedbacks between genotype, its encoded phenotype, and the environment that it must fit. B: Summary of model components. +ve environmental factors are one that must match the expressed phenotype, -ve environmental factors ones which must not match the expressed phenotype. See text for details.

sort of as the *genotype*. The strings are made up of different *characters;* there can be any number of types of characters (if the strings were to mimic DNA or RNA, the number of character types would be 4; the model was run with the number of character types ranging from 2 to 16). There is no equivalent of protein translation in the system. The model consisted of a number of *organisms*–in this initial implementation there were only 5 organisms for computational reasons. The organisms exist in an *environment*. Each organism contains a number of *genes* which together comprise its *genotype*; in the runs reported here, organisms contained 25, 50 or 100 genes. Each gene is composed of up to ten *positive regulatory elements*, up to ten *negative regulatory elements*, and a *coding sequence*. The sum of the coding regions of genes that are active at any one time comprise the organism's *phenotype*. The organism's fitness is the match between its *phenotype* and its *environment* as follows. The *environment* comprises *positive* elements, *negative* elements, and *signalling elements*. Fitness is the sum of the number of *positive environmental elements* that match the current *phenotype* minus the number of *negative environmental elements* that match the current *phenotype*. (This is to reflect that sometimes having

a function in a cell can be detrimental to the cell; were this not true, in our model all cells would express all genes all the time for maximum fitness.)

Gene expression is controlled as follows. A *regulatory element* is active when either it matches an *environmental signalling element* or it matches the *phenotype*. This represents the transduction of an environmental signal into gene activity, and the transduction of internal gene activity into gene activity. If the sum of the number of active *positive regulatory elements* exceed the number of active *negative regulatory elements* then the gene it transcribed and its *coding sequence* is added to the *phenotype*.

The model is seeded with random strings. At each cycle a new *phenotype* is computed, and new fitness computed for each *organism*, and the most fit organism randomly replaces one of the other organisms (which can include self-replacement). The organisms are then mutated by making small, random changes (character changes, insertions or deletions, with a bias of 6:4 deletion over insertion) to a fraction (typically between $10^{-5}$ and $5 \times 10^{-5}$) of the strings in the *genotype*, or completely deleting one of them (typically with a probability between $10^{-6}$ and $5 \cdot 10^{-6}$).

The model components are summarised in Fig 2B

## 2.3 Implementation and availability

The first implementation of the model was done in Excel 2010, and is provided as S2 File to the paper.

## 3 Results

### 3.1 Modelling selection and adaptation

We begin by showing that the model produces results that are consistent with adaptation, i.e. with changing from an initial random state to a state where the average fitness of the organisms is greater than it was at the start. We emphasise that changes made to the components of the model are entirely random; there is no directionality in the model except a slight bias towards gene shrinkage noted below. Both the initial genome and the environmental factors that the genome has to adapt to are randomly generated as well. Adaptation is therefore the result of selection for better 'fitness'.

The environment to which the organisms can adapt can be varied in two ways. The environment to which the population adapts can be static, or dynamic, and it can have differing levels of complexity. Both have parallels in biology, although they are not meant to represent specific scenarios. Increasing the complexity of the environment represents more constraints on the organism, as might be represented by a complex ecosystem such as a rainforest compared to a farm monoculture. Dynamic environments swap between one or more environments, with the organism having to detect which environment it is being presented with and express a gene set appropriate to that environment. The differing environments can also be of greater or lesser complexity. Dynamically changing the environment to which the organisms have to adapt might be represented by changing seasons or the changing environment in the intertidal zone of the seashore. Both types of environmental change were included to explore if selection of genetic control style were different for dynamic vs static environmental complexity.

We can measure the 'degree of perfection' $P$ of an organism in terms of the evolved fitness $F$ as a fraction of the possible maximum fitness, as the maximum fitness is the number of environmental factors $E_f$. Some example fitness curves are shown in Fig 3. Fig 3A shows a typical curve that reaches a plateau of fitness and then does not achieve any greater fitness in the run. Fig 3B shows a curve that is similar to ~500,000 generations, but then a new increase in fitness

is observed. Fig 3C shows an example where the organism was adapting to three environments which varied with time. Fig 3C shows the decomposition of the fitnesses to each of three environments in a model, together with the average across all environments. In this run, the organism is tested against one of three, unrelated environments; the environment that the organism has to match changes every two generations. Note that fitness to each environment does not increase in parallel–sometimes selection has resulted in better fitness for one environment, sometimes for another. Fitness for an environment can actually decline if overall fitness does not decrease substantially. Fig 3D shows the separate fitness trajectories of five organisms as

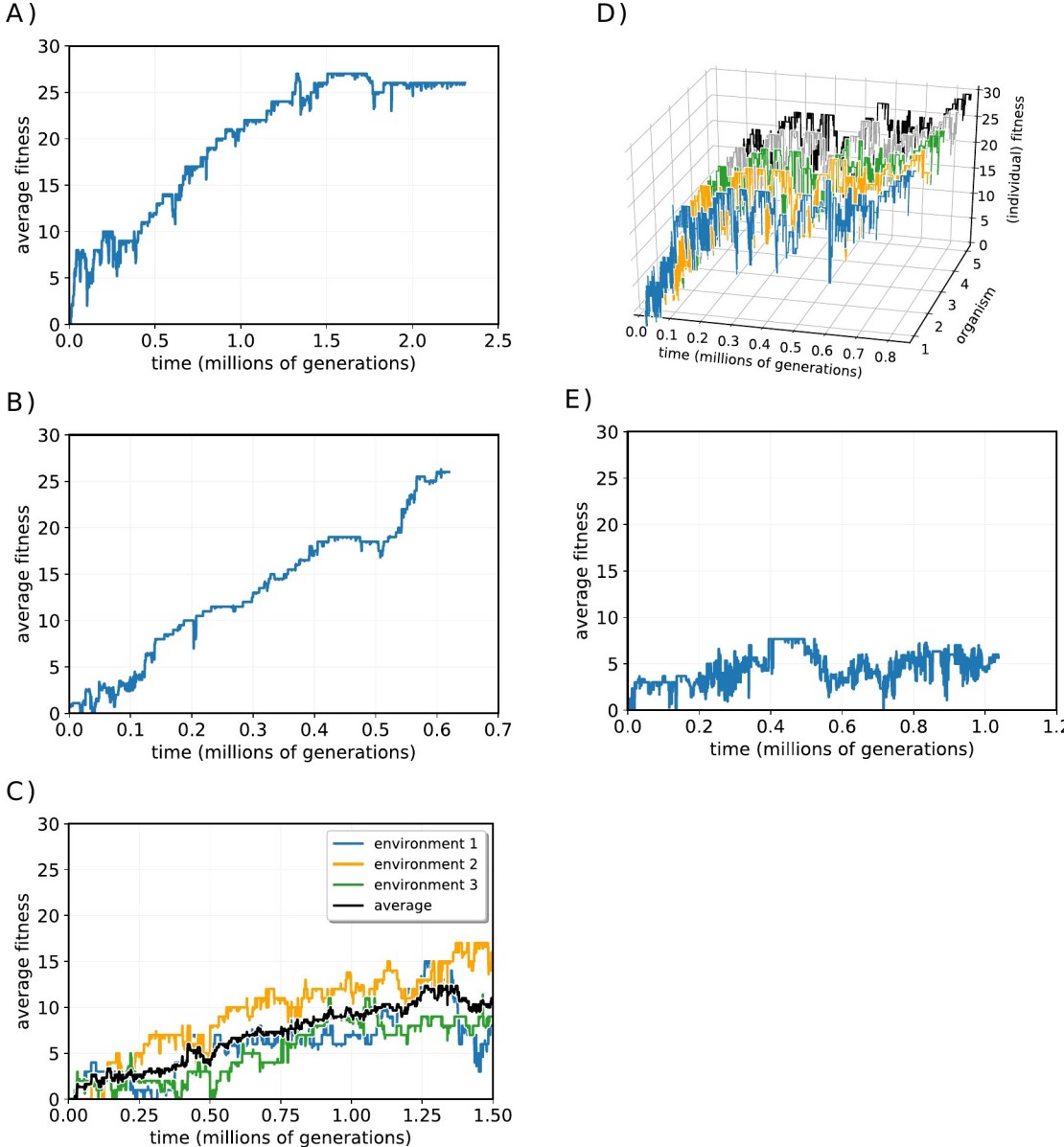

**Fig 3. Examples of fitness plots for different runs of the model.** For all curves: Y axis = fitness, X axis = time in units of generations. A: Average fitness of the population converges smoothly on a maximum. B. Average fitness shows a jump in adaptation at 500,000 generations. C. convergence of average fitness across three environments, showing divergent adaptation to each of the environments. D. Fitness of each of the five organisms making up the population plotted separately in a run that converges on a solution. E. Plot of fitness in a run that fails to converge on an optimum fitness.

they evolve in a single environment. Again, individual organism can lose fitness, but the population trend is usually to increasing fitness. Lastly, Fig 3E shows a model that has not evolved significantly. Most of the change in fitness in Fig 3E appear to be noise, and fitness wanders around a low average ($P{\sim}0.04$ in this case, as $E_f = 100$).

### 3.2 Failure to adapt

Models did not converge onto a fit state $\sim{}^1/_5$ of the time (depending on what 'fit' means, and the selection of parameters). This was found to be a function of the degree to which the genome complexity can match the environmental complexity (Fig 4). Highly complex environments are not efficiently matched by low complexity genomes, as would be expected. This to a degree is a consequence of limited run-time: some model runs had low fitness for a time and then experienced a 'jump' in fitness as a low-probability solution was 'discovered' (e.g. Fig 3B).

We define whether a population is converging on a solution with a Curve Parameter $C_p$ as follows: we define a time $P$ as the time at which the population reaches a plateau of adaptation, i.e. does not appear by inspection to be able to increase its adaptation. $C_p$ distinguishes between populations that smoothly approach such a fitness plateau, such as shown in Fig 3A, and populations whose fitness fluctuates, such as in Fig 3E. Thus, if the fitness at times $0.25{\cdot}P$,

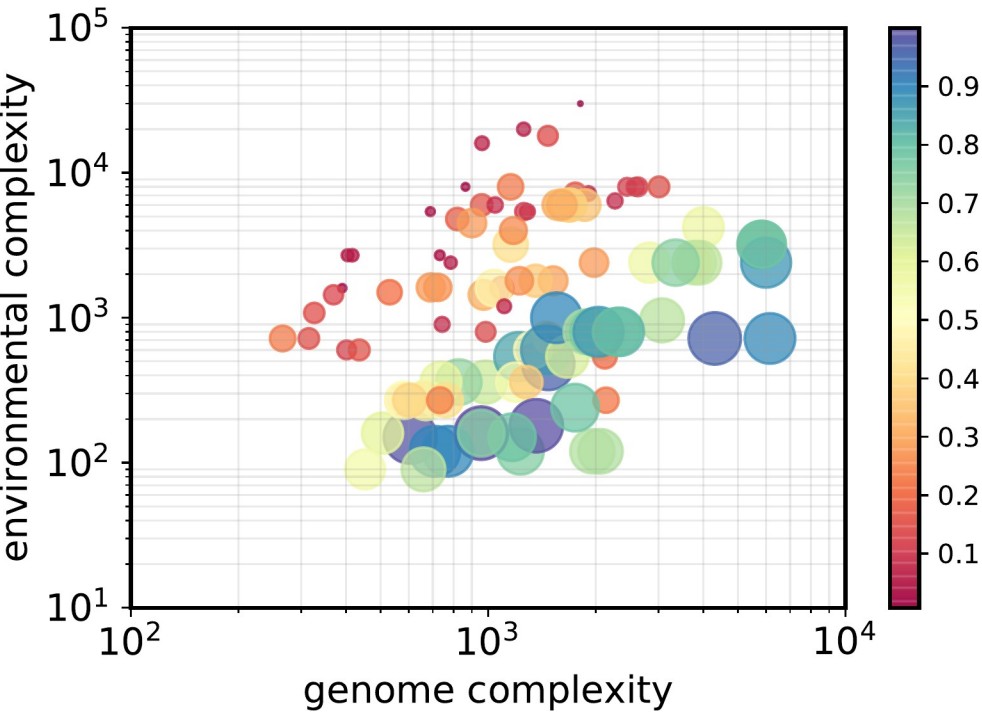

**Fig 4. Selection is inefficient for runs with a combination of high environmental complexity and low genome complexity.** X axis: Genome complexity (number of genes in each organism time the number of types of bases of which those genes are made up, times the average length of the genes at the end of the selection process. Y axis: Environmental complexity; number of environmental factors that must match the genotype, multiplied by the number of different environments to be fitted, the number of base types in each sequence, and the length of the environmental strings to be matched. The 'perfection index'–fitness at selection plateau divided by maximum possible fitness–is both proportional to the circle areas and, for enhanced readability, to the colour scale (vertical bar).

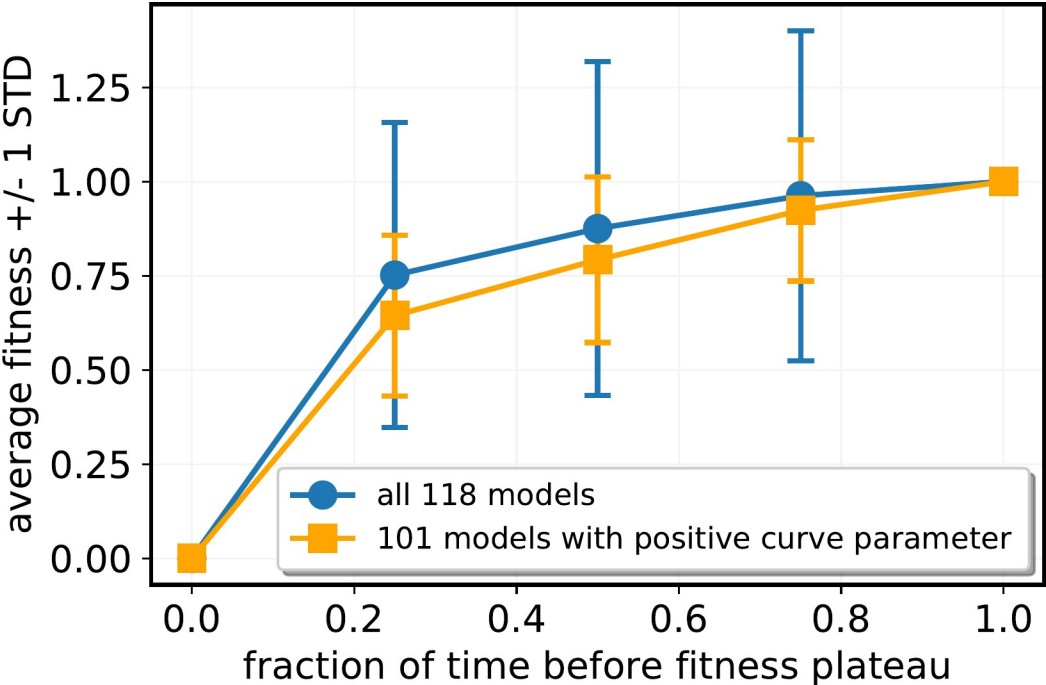

**Fig 5. Average fitness of the model runs.** Curves are normalized to maximum fitness = 1. The X axis shows the fraction of time until the fitness reaches a stable plateau. The average fitness across all 118 model runs was calculated for time = 0, time = between 0.25 and 0.5, time = 0.5–0.75, time = 0.75–1 and time>1 (i.e. on the plateau of fitness). By definition, fitness = 1 when time>1. Error bars are standard deviation. Blue curve: All 118 model runs. Orange curve: 101 model runs for which Curve Parameter $C_p$>0.

0.5·$P$, 0.75·$P$ and $P$ are A, B, C, D respectively, and S(x) is the sign of x, (such that x>0 $\Rightarrow$ s(x) = 1; X<0 $\Rightarrow$ s(x) = -1; x = 0 $\Rightarrow$ s(x) = 0) then $C_p$ = s(B-A)+s(C-A)+s(D-A)+s(C-B)+s(D-B)+s(D-C).

If A<B<C<D (i.e. fitness is increasing throughout the run), then $C_p$ = 6

For this analysis, runs of the model were only used if the curve parameter $C_p$ is greater than zero. 101 out of 118 runs of the model met this criterion. Omitting curves for which $C_p \leq 0$ resulted in substantially less scatter in the adaptation curves averaged across all models (Fig 5).

Would all models converge on an optimal solution eventually? We hypothesise they would, but it might take years to achieve this using the relatively inefficient coding, which for practical reasons was run for only an average of $1.4 \cdot 10^6$ generations. (For comparison, the long-term evolution experiments performed by the Lenski lab. have been running for more than 20,000 generations, and show a range of adaptations in gene control structure without changing the underlying mechanisms or logic of the gene control architecture [76–78]). Models were therefore stopped when they seemed to have reached a steady state of control structure (as defined below) and fitness within this time window. Future work will seek a more objective measure of termination state through spectral analysis of the fitness functions in an exhaustive scan of our parameter space. Disentangling the typical timescale of purely stochastic fluctuations from the timescale of selection-induced changes will provide a useful estimate of the average, expected time until convergence, as well as estimates of the range on that time.

### 3.3 Coding region evolution

There is a slight bias the mutation mechanism towards gene shrinkage, included because a) this is seen in real mutation rates and b) it protects the model against indefinite expansion of

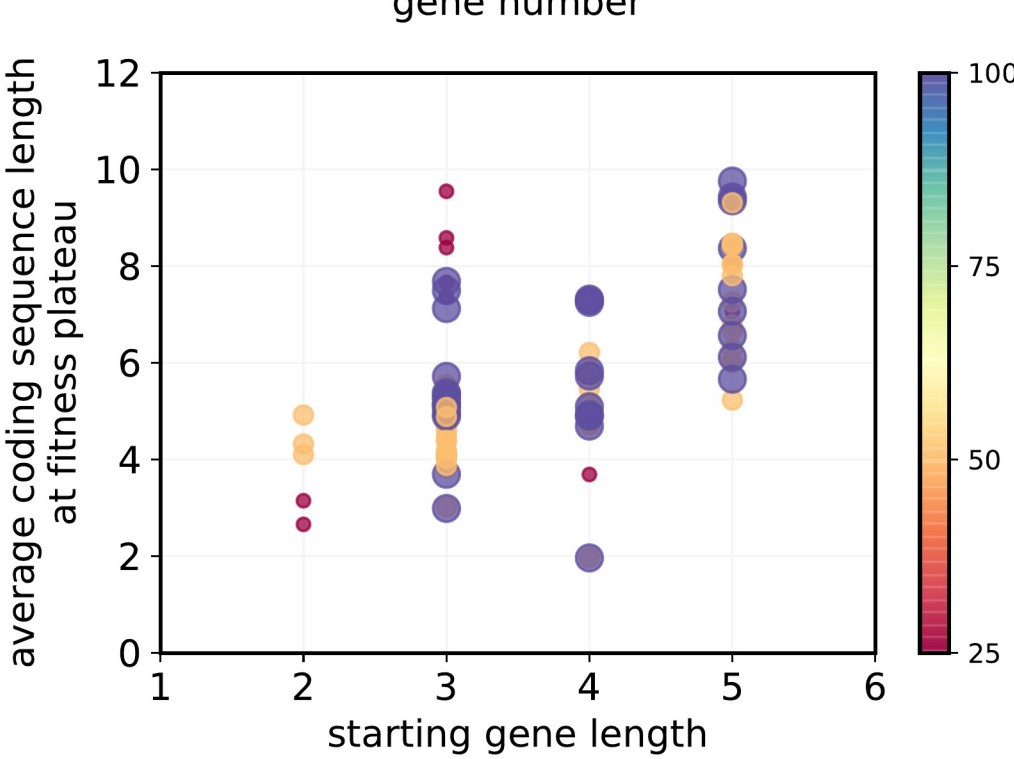

**Fig 6. Length of the coding sequences averaged across all genes in a run (Y axis) as a function of the starting length of those sequences (X axis).** Circle sizes and color scale show the number of genes in the run, and show no distinct pattern.

genes through 'drift'. Despite this, the average length of coding regions tends to increase with model progression (Fig 6). This is explicable as follows. Gene activation depends on matching part of an expressed coding sequence to a regulatory sequence. Thus larger genes mean a greater chance of productive interaction with a regulatory element. The only selective pressure against long genes is the chance that they interact with one of the 'negative' environmental elements when they are expressed. As for all runs there are more regulatory elements (20 per gene) than environmental factors (a maximum of 200), this provides a selection pressure towards longer genes.

## 3.4 Default genetic control measures

The computational effort to exhaustively analyse the entire control network of up to 100 genes each interacting with up to 20 control elements in each gene in each of 5 organisms in 100 runs or up to 50,000 timesteps each is unrealistic, and so we summarise the overall style of control as follows. We use two types of measure of control style; 'new gene' and 'existing genome' measures.

'New gene' measures measure statistically whether a new, random gene inserted into the genome is likely to be expressed or not. This can be estimated by statistics on the length of regulatory elements. A short regulatory element is more likely to match a new (random) sequence in the phenotype than a long regulatory element, because a short string is more likely to match a random target by chance than is a long string. (Consider the chance that the strings "A" and "ALPHABET" will match the text in this paper) Thus if negative regulatory elements are on

average shorter than positive regulatory elements in a gene, it is likely that the gene is not active. Thus the ratio

$$R_a = \frac{\frac{\sum negative\ element\ length}{\sum negative\ element\ count}}{\frac{\sum positive\ element\ length}{\sum positive\ element\ count}}$$

Is a measure of the relative probability that negative elements matched a random sequence vs positive elements, and thus of 'default' regulatory style such that a lower $R_a$ implies a bias towards a 'default off' regulatory style. (Zero-length elements, i.e. ones which have been deleted, are not counted in the average).

The same argument means that the shortest regulatory element in a gene is the one most likely to be 'active' in a gene. If the shortest regulatory element is positive, then there is a greater chance that the gene will be active; if negative, then the gene is more likely to be inactive. We therefore also adopted a measure looking for the shortest regulatory element in a gene. For each gene, the shortest non-zero control sequence is recorded for positive and negative control elements. The average of the length of the shortest regulatory element for all genes in the genotype is reported. Thus the ratio

$$R_m = \frac{\frac{\sum min.negative\ element\ length/gene}{\sum genes}}{\frac{\sum .min.positive\ element\ length/gene}{\sum genes}}$$

Is a measure of the bias in regulatory style, such that a lower value implies a 'default off' regulatory style. (Again, zero-length elements, i.e. ones which have been deleted, are not counted in the average)

'Existing genome' measures examined the potential regulatory circuitry in the adapted genome, and is an analogy for the existence of specific pairs of regulator+sequence in the control systems of a cell. Again, two measures were used. The first is the ratio of positive to negative regulatory elements that match a coding sequence in the genome (i.e. elements that could be active).

$$E_a = \frac{\sum(positive\ regulatory\ elements\ with\ matches\ in\ coding\ regions)}{\sum(negative\ regulatory\ elements\ with\ macthes\ in\ coding\ regions)}$$

A gene only has to have more active positive than negative regulatory elements to be active. $E_a$ could therefore give a false view if a small number of genes had a large preponderance of potentially active regulatory elements and a larger number of genes had a small preponderance of potentially active negative elements. A second 'existing genome' measure therefore looked at potentially active elements on a per-gene basis, by averaging the fraction of potentially active regulatory elements (i.e. ones that matched a coding region in the genome) as a fraction of all potentially active regulatory elements

$$E_g = 2 * average\left[\frac{(positive\ reg.elem.mtching\ a\ coding\ region\ per\ gene)}{(any\ reg.elem.mtching\ a\ coding\ region\ per\ gene)}\right]$$

$E_g$ is multiplied by, so that for all measures a value of less than 1 suggests a 'default off' mode, a value of greater than 1 suggests a 'default on' mode.

The reader may be puzzled that the 'new gene' measures divide negative element lengths by positive element lengths, which the 'existing gene' measures divide positive element numbers

by negative element numbers. This is because the 'new gene' measures are based on values that reflect the chance that a regulatory element matches a random, unknown sequence. The probability p that a random sequence of N characters (where there are C choices of characters) matches another random sequence of N characters is given by

$$p = \frac{1}{C^N}$$

In other words, the negative element 'new gene' measures are inversely related to the probability that the negative element is active, and conversely the positive gene measures are inversely related to the probability that the positive element is active. Thus $R_a$ and $R_m$ are larger if there is a greater chance that a positive element is active. The 'existing gene' measures, however directly relate to which element is active, it directly probes which elements can be active in the genome, regardless of their size. So, again, for $E_a$ and $E_g$ are larger if there are more positive elements potentially active.

Correlations of these two measures to both the inputs and the outputs of model runs are provided in Table 1. We emphasise that this is an initial modelling study, and much more extensive modelling with more efficiently coded models and better hardware will be needed to

**Table 1. Correlations with control logic style.**

| Model parameters | Measures of genetic control logic ($<1$ = 'default off') | | | |
|---|---|---|---|---|
| | 'New gene' measures | | 'Existing genome' measures | |
| | Rm | Ra | $E_a$ | $E_g$ |
| *Starting parameters* | | | | |
| Number of environments | -0.02 | 0.032 | -0.096 | -0.091 |
| Env, Factors (N = 1) (a) | 0.102 | 0.08 | 0.001 | -0.014 |
| Env. Factors (N>1) (a) | 0.419** | 0.24 | 0.358* | 0.302* |
| Length of initial gene | 0.445 ** | -0.044 | 0.24* | 0.315* |
| Number of genes | -0.144 | -0.144 | -0.236* | -0.223* |
| Genome complexity at start (b) | 0.297 ** | 0.096 | 0.184 | 0.202 |
| *Parameters at fitness plateau* | | | | |
| Adaptation at plateau | 0.326 ** | 0.167 | 0.267* | 0.288** |
| Fraction of perfection | 0.117 | -0.015 | 0.197 | 0.183 |
| Average gene length | 0.683 **** | 0.198 | 0.508*** | 0.529**** |
| Genome complexity at end (b) | 0.373 ** | 0.18 | 0.258* | 0.254* |
| regulatory complexity at end (c) | 0.13 | 0.184 | 0.134 | 0.135 |
| Number of genes expressed | 0.262 * | 0.102 | 0.125 | 0.149 |
| Fraction of genes expressed | 0.713 **** | 0.454 ** | 0.577**** | 0.616**** |

Correlations between two measures of genetic 'style' and some inputs and outputs from models. (a) Number of environmental factors that the model must adapt to, separated into runs where fitness if determined for only one environment (N = 1) or more than one environment (N>1). (b) Genome complexity = [number of characters]*[number of genes]*[average length of coding regions]. (c) Regulatory complexity = [number of characters]*[number of genes]*[average length of regulatory elements]. _ "Min–ve" = average length of the shortest negative regulatory element in each gene, averaged over all genes. "min +ve"= average length of the shortest positive regulatory element in each gene, averaged over all genes. "Avg–ve" = average length of all non-zero-length negative regulatory elements in the genome."Avg +ve" = average length of all non-zero-length positive regulatory elements in the genome. Significance of the correlations (i.e. chance that the observed correlation is seen in 101 model runs if the measures of control logic are not correlated with the input parameters)

* = p<0.05

** = p<0.01

*** = p<0.001

**** = p<0.0001.

confirm, expand and dissect these findings. However three patterns are clear from the results here.

Correlations are notably weaker for $R_a$ (the ratio of negative average regulatory element to positive regulatory element length). This may be related to a slight bias in the evolution of $R_a$, observed in Section 3.6 below, which introduces additional noise into the model. However we include this result here for completeness.

Firstly, for static environments the complexity of the environment has no effect on whether default on or default off genetics evolves. This was a surprising result, as we expected more complex environments to drive selection for more complex genetic controls, with consequences (positive or negative) for a 'default off' control style. For fluctuating environments, more complex environments were weakly correlated with some measures of default on genetics. However the effects are weak (these significance levels are not Bonferroni corrected). The lack of obvious environmental effect may relate to the duration of the modelling, as noted in section 3.2 above; simple genomes could not adapt to complex environments in the time available. The weakness of environmental effects may also be an artefact of the small sizes of the populations (leading to noise which obscures patterns), the small number of combinations of parameters explored (61 out of 720 possible combinations of the parameters used in these runs), or the small genomes (maximum 100 genes).

Secondly, genome complexity both at the start and at the end of the models is correlated with 'default on' control style (i.e. positive correlation with all the measures of genetic logic, for which higher values mean a more 'default positive' control logic). Again, this was a surprise. Our hypothesis is that 'default off' genetics allows complex genome evolution. However, our initial hypothesis, that 'default off' genetics allows ready gene duplication, is not captured in this model, where the number of genes is fixed.

Lastly, 'default on' genetics is most strongly correlated with the number of expressed genes. We further dissect this in Fig 7. There is a striking correlation between two measures of 'default control logic' and the number of expressed genes. If relatively few genes are expressed in a genome, then 'default off' is preferred. If many genes are expressed, 'default on' is preferred. Note that the correlation with the number of expressed genes is much weaker (Table 1)–it is the fraction of the genome which is expressed that correlates more strongly with genetic logic than any other parameter. We note again that 'default off' does not dictate how many genes are expressed,

## 3.5 Reproducibility

No stochastic model will give the same results in different runs, so it is important to show that the variability in output is not so extreme as to render results uninterpretable. The purpose of this modelling was to test the model concept and provide an initial exploration of parameter space: as a result, only a few sets of runs of the model were replicate runs with the same parameters. We chose three runs that gave different control logic outputs in an initial run and re-ran the same parameters with different starting genomes and environments. The results are summarised in Fig 8. This shows that, while results are variable, the genetic control outputs, and specifically the $R_m$ parameter, are consistent within replicates: Replicates of a model run that gave $R_m = 1$ consistently gave $R_m \sim 1$ ("Replicate 1), Replicates of a run which gave $R_m > 1$ ("Replicate 2") consistently gave $R_m > 1$, and Replicates of a run that gave $R_m < 1$ consistently gave $R_m < 1$ ("Replicate 3").

## 3.6 Adaptation is the result of selection

As this model is complex and produces large amounts of complex data, it is important to show that results are due to selection and not to inherent biases in the model. We therefore re-ran

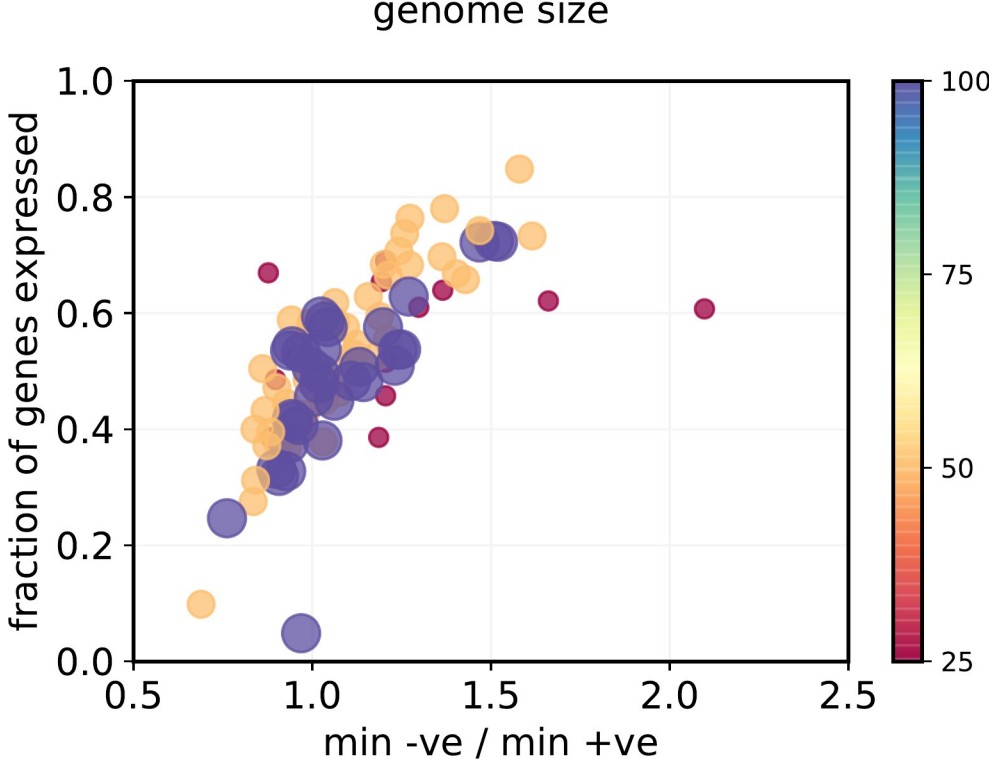

**Fig 7. Relationship between the ratio of minimum negative elements to minimum positive elements (X axis: <1 = 'default off') to the fraction of genes in a genome expressed at fitness plateau (Y axis).** Both circle area and color scale are proportional to genome size (25, 50 or 100 genes). "Min–ve" = average length of the shortest negative regulatory element in each gene, averaged over all genes. "min +ve"= average length of the shortest positive regulatory element in each gene, averaged over all genes.

the 'Set 2' runs above but with organisms selected at random rather than on the basis of fitness. (This required a single cell change in the model.) Replicate 2 converged within $10^6$ steps, and so the model was run to $1.5 \cdot 10^6$ steps to ensure compatibility. 8 runs were completed. The results are summarised in Table 2. None showed any significant net adaptation or fitness. One measure of default genetic control ($R_a$, the ratio of average positive regulatory length to negative regulatory element length) showed a slight bias towards a value <0; the source of this bias has not been determined, but is probably related to the residual biases in Excel's random number function; this could be tested in the future by replacing this function with a truly non-repeating source of numbers, such as the digits of Pi. $R_a$ was the measure with the weakest correlations to any outcome in Table 1. There is also a clear difference between runs with selection and runs without for all measures that $R_m$. The range across all the model runs greatly exceeds the range seen with no selection, showing that despite small biases in 'mutation' selective effects dominate changes in the output of the model

## 4 Discussion and conclusions

We have presented a model of the evolution of genome control logic, and an initial analysis of its performance on a small number of test cases. The model performs in a comprehensible way, and evolves fitter organisms. Preliminary statistics suggest that the model performance is stable, i.e. a given set of starting conditions will give a set of outputs more closely related to each other than random, despite the model being a stochastic one.

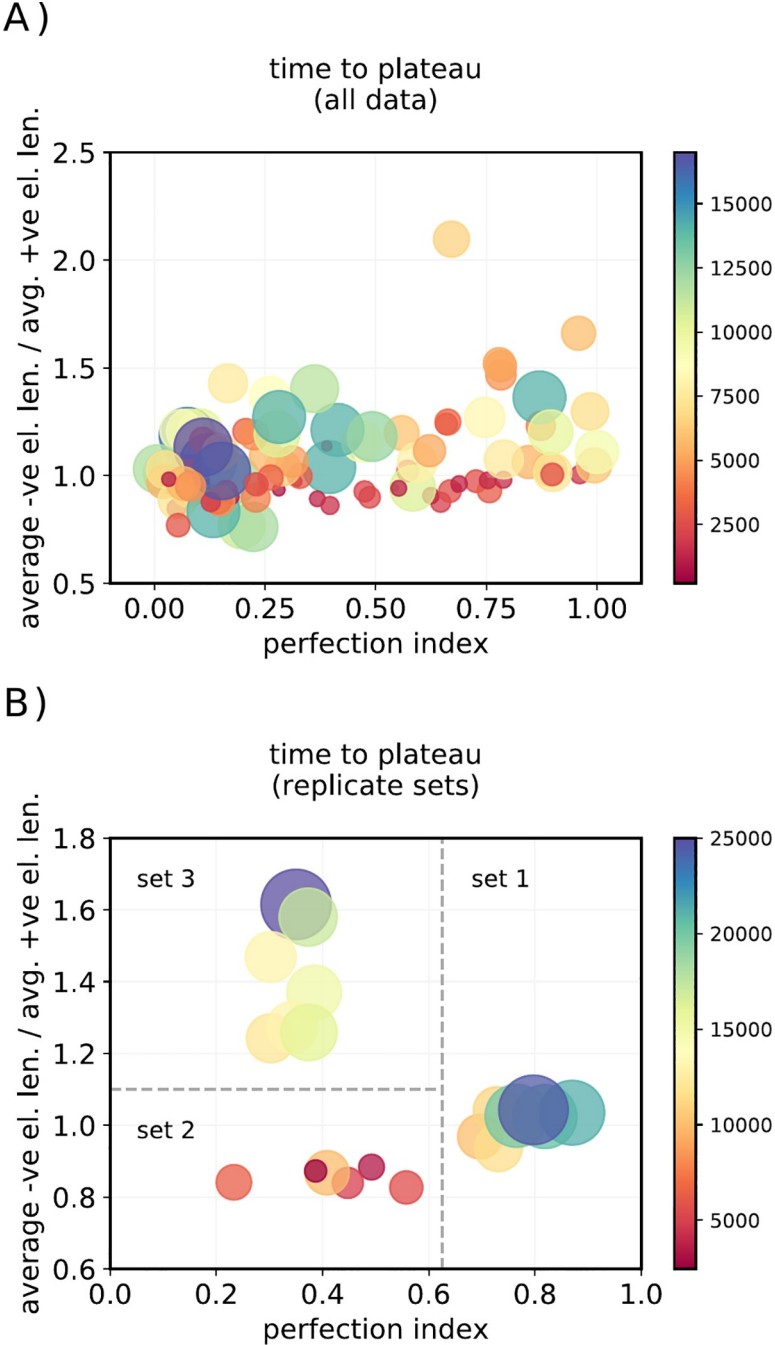

**Fig 8. Reproducibility across runs.** A: Example outcomes from all runs with diverse starting conditions, and B: From replicate sets of runs started from the same set of parameters. X axis: 'Perfection index' (fitness at the fitness plateau as a fraction of the maximum possible fitness with those parameters). Y axis: Ratio of the length of the average minimum negative regulatory element length to the average minimum positive regulatory element length. Both circle area and color scale are proportional to the number of steps taken to reach a fitness plateau.

We emphasise that this is a preliminary exploration of this model only, and much more needs to be done. However with that caveat, the results show three things of potential interest to the hypothesis that stimulated its creation

**Table 2. Results from non-selected model runs.**

| | Number of model runs | Fitness | $R_a$ | $R_m$ | $E_a$ | $E_g$ |
|---|---|---|---|---|---|---|
| Replicate series 2 | 6 | 12.578 (1.322) | 1.0064 (0.0098) | 0.8685 (0.0103) | 0.8385 (0.0286) | 0.9091 (0.0238) |
| Non-selected control runs | 8 | 0.344 (0.0857) | 0.9696 (0.0071) | 0.9888 (0.0145) | 0.9966 (0.0666) | 1.0097 (0.0288) |
| Range in all selected runs | - | 0.7452–114.14 | 0.6294–1.2573 | 0.6889–2.0972 | 0.7634–2.8182 | 0.8401–1.5848 |

Comparison of results of replicate set 2 (from Fig 8) with the models run with the same parameters but with replication of organisms uncoupled from fitness. Values in brackets are standard error of the mean.

i. The genetic logic a population of organisms evolves is only weakly related to the complexity of the environment it finds itself in. This was unexpected.

ii. The evolved genetic logic is strongly related to the starting and the final, evolved genome complexity. More complex genomes have 'default on' logic. This is not predicted by the model, but as the model's predictions on evolution of genetic logic refers primarily to the acquisition of new genes in the genome, an aspect of evolution not captured here, this does not test the hypothesis.

iii. The strongest correlations with genetic logic are with the fraction of the genome that is expressed.

Point (iii) above fits with (although is a weak test of) our original hypothesis. It also fills in a significant gap in the hypothesis about why a 'default off' logic should be selected. Clearly, an organism cannot evolve 'default off' in anticipation of acquiring new genes. However if a specific combination of environmental and genetic features encouraged the development of 'default off' genetics, then such an organism would be pre-adapted for genome complexification by gene duplication and divergence. As noted in the introduction, the majority of genes in eukaryotic genomes are not expressed at any one time. Most of them are 'off'. Our model appears to be evolving a similar expression pattern in some cases, and in those cases the 'default off' logic is selected.

If our results represent the more complex world of real genetics, then we might speculate that organisms living in an environment that occasionally called on a diverse set of genes but most of the time did not require them would feel short-term selective pressure to evolve a 'default off' logic. Such an environment could be one in which a heterotroph lived in a community made up of a changing composition of autotrophs, each of which provided a small number of substrates to the heterotroph. If such a scenario were valid, then we would expect more comprehensive modelling to reveal influences of environmental change on both expressed gene numbers and default logic. Such work is being actively pursued.

## Supporting information

**S1 File.**
(PDF)

**S2 File. Evolution model—spreadsheet version 7.**
(XLSM)

## Acknowledgments

We are grateful to our anonymous reviewer, who suggested several improvements to the paper and additional analyses that improved this paper substantially.

## Author Contributions

**Conceptualization:** William Bains, Dirk Schulze-Makuch.

**Investigation:** William Bains, Enrico Borriello.

**Methodology:** William Bains.

**Software:** William Bains.

**Visualization:** Enrico Borriello.

**Writing – original draft:** William Bains, Dirk Schulze-Makuch.

**Writing – review & editing:** William Bains, Enrico Borriello, Dirk Schulze-Makuch.

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
