## [Decision Letter · Decision Letter 0]

18 Feb 2021

PONE-D-21-00488

Evolution of default genetic control mechanisms

PLOS ONE

Dear Dr. Bains,

Thank you for submitting your manuscript to PLOS ONE. After careful consideration, we feel that it has merit but does not fully meet PLOS ONE’s publication criteria as it currently stands. Therefore, we invite you to submit a revised version of the manuscript that addresses the points raised during the review process.

We look forward to receiving your revised manuscript.

Kind regards,

Ernesto Perez-Rueda

Academic Editor

PLOS ONE

Journal Requirements:

2. Please include captions for your Supporting Information files at the end of your manuscript, and update any in-text citations to match accordingly. Please see our Supporting Information guidelines for more information: http://journals.plos.org/plosone/s/supporting-information

Reviewers' comments:

Reviewer's Responses to Questions

**Comments to the Author**

1. Is the manuscript technically sound, and do the data support the conclusions?

Reviewer #1: No

2. Has the statistical analysis been performed appropriately and rigorously? 

Reviewer #1: No

3. Have the authors made all data underlying the findings in their manuscript fully available?

Reviewer #1: Yes

4. Is the manuscript presented in an intelligible fashion and written in standard English?

Reviewer #1: No

5. Review Comments to the Author

Reviewer #1: In this manuscript, motivated by the switch from “default on” to “default off” logic during the eukaryogenesis, the authors built a model to simulate the evolution of the default mode of gene expression and test the correlation between the mode and various model parameters. While I appreciate the authors’ efforts, I have several doubts on different components of the simulation, which makes me question how much the simulation can tell us on the biological questions.

First of all, there is probably a problem in the fundamental definition of “default off” logic. In Line 115, the authors stated, “We argue that this ‘Default off’ logic is more efficient if the majority of the genome is silent.” Is this tautology? I think “the majority of the genome is silent“ is just synonymous with the “default off” logic.

Second, the assumptions about “environments” is not very clear. And the underlying biological reasoning is not clear as well. In Fig 3A-B, the authors demonstrated a simple case under a certain environment. Then suddenly in Fig 3C, there are three environments. Both the purpose of increasing the number of environments and the biological scenarios to depict by the increase are not very clear. Especially, the author attempted to test the hypothesis on the environmental complexity and the evolution of the default mode. Presumably, the optimal goal of gene control is to express different certain sets of genes at different time or at different habitats. That could be modeled by fluctuations with these “environments.” But in this study, it seems that the environmental complexity only represents the number of possible environments existed for matching instead of frequent alternations among possible environments with time. I am really curious if using the fluctuating environments would lead to different conclusions.

Also, I have doubts on the measurements for “default off” and “default on” logic. While I understand why the shorter elements have a higher likelihood to be active in the model, that does not seem to reflect the real biology of gene regulation/control. The activity of elements is usually determined by the matching specificity between the sequence of regulatory elements and the regulatory molecules in the environments. It is not easy to rationale the biological meaning on the role of length in the simulation here. After all, I feel the evolution of "default off" logic results from the gains of negative elements with strong specificity instead of the decreasing specificity of negative elements for spurious interactions. Of course, whether this is true requires more empirical data.

Finally, the author stated they found a “striking” correlation between the fraction of expressed genes and the tendency of “default on” logic. But if the “default on” logic suggests that more positive elements are active, isn’t the higher fraction of expressed genes just a natural outcome of that?

Minor comments:

In Fig 2a, why the arrow of transcription is from the phenotype to genotype? It should be opposite.

In Fig 2b, what “ve” means should be explained in more details in the text around page 6.

There is an error on reference format in Line 193.

I appreciate the authors have considered the percentage to full adaptation and illustrated cases of adaptation very clearly. But I would further encourage the authors to also run the simulation without any fitness differences in the model at all and see if non-adaptation parameters in the models (e.g. mutation rates and choice of measurements) are sufficient to create the correlations (a negative control of the methodology).

6. PLOS authors have the option to publish the peer review history of their article (what does this mean?). If published, this will include your full peer review and any attached files.

Reviewer #1: No

---

## [Author Response · Author response to Decision Letter 0]

29 Mar 2021

[Auth] We are grateful to the reviewer for their detailed comments. We provide a formatted version of our response uploaded with the MSS. In that, the reviewer's comments are in bold, our responses are in plain test. Here we flag our responses with [Auth]. Line numbers refer to the Track Changes version of the revised paper.

First of all, there is probably a problem in the fundamental definition of “default off” logic. In Line 115, the authors stated, “We argue that this ‘Default off’ logic is more efficient if the majority of the genome is silent.” Is this tautology? I think “the majority of the genome is silent“ is just synonymous with the “default off” logic.

[Auth] We apologise for not making this clearer. The concept of a ‘default off’ logic is that the genome controls are structured such that it is ‘easier’ for a gene to be inactive than to be active. Specifically, more metabolic energy needs to be expended to turn a gene on in a ‘default off’ genome. This does not mean that genes are inactive. The entire genome could be active at the same time in a ‘default off’ genome – that would just take a lot more metabolic energy to achieve than in a ‘default on’ genome. 

[Auth] To use an analogy, it requires less energy for a human to be sitting down than running. Our ‘default’ state is sitting down and (relatively) inactive. That is not to say that marathons, where everyone is running, cannot happen, just that they take energy to make happen. 

[Auth] We have made this point explicitly in lines 118 – 124., and added some additional text to line 97 – 99 and 396

Second, the assumptions about “environments” is not very clear. And the underlying biological reasoning is not clear as well. In Fig 3A-B, the authors demonstrated a simple case under a certain environment. Then suddenly in Fig 3C, there are three environments. Both the purpose of increasing the number of environments and the biological scenarios to depict by the increase are not very clear.

[Auth] We have revised the text around this substantially, adding new explanation in lines 204 - 215. In essence, the complexity of the environment to which a genotype adapts could be a single complex environment (say, a rainforest) or several simpler environments which alternate (say, a tidal pool at high and low tide). We have tried to make the rationale for including both in our model clearer. 

[Auth] Figure 3C was included to illustrate that the model does adapt when presented with multiple alternating environments, but if the reviewer thinks this is confusing and does not help understanding, we can remove this illustration. 

Especially, the author attempted to test the hypothesis on the environmental complexity and the evolution of the default mode. Presumably, the optimal goal of gene control is to express different certain sets of genes at different time or at different habitats. 

[Auth] This is our expectation, yes, and the model is built around this idea

That could be modeled by fluctuations with these “environments.” But in this study, it seems that the environmental complexity only represents the number of possible environments existed for matching instead of frequent alternations among possible environments with time. I am really curious if using the fluctuating environments would lead to different conclusions.

[Auth] We also were interested in this! We have expanded our discussion of the environmental parameters in lines 202 – 213 and 375 - 380. We have also addressed this with some additional analysis, which shows that whether the environment fluctuates or not does affect the correlation of environment with default genetic mode. However it suggests that fluctuating environments are weakly correlated with ‘default on’ genetics, which is not what we would expect. We discuss this briefly (lines 369 – 373). We have not explored enough of the parameter space to see if the rate of fluctuation or the number of different environments has an effect. 

Also, I have doubts on the measurements for “default off” and “default on” logic. While I understand why the shorter elements have a higher likelihood to be active in the model, that does not seem to reflect the real biology of gene regulation/control. The activity of elements is usually determined by the matching specificity between the sequence of regulatory elements and the regulatory molecules in the environments. It is not easy to rationale the biological meaning on the role of length in the simulation here. 

After all, I feel the evolution of "default off" logic results from the gains of negative elements with strong specificity instead of the decreasing specificity of negative elements for spurious interactions. Of course, whether this is true requires more empirical data.

[Auth] The reviewer brings up a very good point. In Eukaryotes controls with very little sequence specificity (analogous to short sequences in our model) are common in the folding and compaction of chromatin, e.g. the binding of H2A/H2B histone pair to nucleosomes, which require energy to remove but are essentially sequence-agnostic. This is accurately captured by our model. However other controls are likely to be sequence specific. 

[Auth] We have therefore re-analysed the model runs to ask whether negative elements that interact with ‘coding’ sequences present in the same cell also show correlations with genome or environmental factors (lines 309 – 312, 336 – 350 and new columns in Table 1). This explicitly probes the regulatory sequence + regulator pair that the reviewer mentions. We have added this analysis to Table 1. The result show very similar correlations to the existing model.

Finally, the author stated they found a “striking” correlation between the fraction of expressed genes and the tendency of “default on” logic. But if the “default on” logic suggests that more positive elements are active, isn’t the higher fraction of expressed genes just a natural outcome of that?

[Auth] As above, ‘default on’ means that it is ‘easier’ to turn a gene on, not that it necessarily is on under any selective regime. We agree that the result is ‘obvious’, but to our knowledge it has not been pointed out before that the nature of the gene control architecture of eukaryotes is best suited to a genome that is mostly not expressed at any one time. 

Minor comments:

In Fig 2a, why the arrow of transcription is from the phenotype to genotype? It should be opposite.

[Auth] Correct – our apologies! We have corrected this. 

In Fig 2b, what “ve” means should be explained in more details in the text around page 6.

[Auth] We have added some text on this, in lines 156 - 157

There is an error on reference format in Line 193.

[Auth] We have corrected this

I appreciate the authors have considered the percentage to full adaptation and illustrated cases of adaptation very clearly. But I would further encourage the authors to also run the simulation without any fitness differences in the model at all and see if non-adaptation parameters in the models (e.g. mutation rates and choice of measurements) are sufficient to create the correlations (a negative control of the methodology).

[Auth] This is a good idea, and we thank the reviewer for it. We have now carried out a set of ‘control’ model run where replication was uncoupled from fitness, and have included details of these in the revised paper, in a new section 3. (lines 423 - 441). This exercise did uncover a bias in one measure, which explains why this measure was more poorly correlated with outcomes.

---

## [Decision Letter · Decision Letter 1]

20 Apr 2021

PONE-D-21-00488R1

Evolution of default genetic control mechanisms

PLOS ONE

Dear Dr. Bains,

Thank you for submitting your manuscript to PLOS ONE. After careful consideration, we feel that it has merit but does not fully meet PLOS ONE’s publication criteria as it currently stands. Therefore, we invite you to submit a revised version of the manuscript that addresses the points raised during the review process.

ACADEMIC EDITOR:

I consider that the manuscript is adequate to be published in PLOS ONE previous to clarify the reviewer concerns associated the section of "Existing genome’ measures". Please, clarify the definition and its consistence witht eh R_a and R_m definitions, and their posterior discussion.

We look forward to receiving your revised manuscript.

Kind regards,

Ernesto Perez-Rueda

Academic Editor

PLOS ONE

Journal Requirements:

Reviewers' comments:

Reviewer's Responses to Questions

**Comments to the Author**

1. If the authors have adequately addressed your comments raised in a previous round of review and you feel that this manuscript is now acceptable for publication, you may indicate that here to bypass the “Comments to the Author” section, enter your conflict of interest statement in the “Confidential to Editor” section, and submit your "Accept" recommendation.

Reviewer #1: (No Response)

2. Is the manuscript technically sound, and do the data support the conclusions?

Reviewer #1: No

3. Has the statistical analysis been performed appropriately and rigorously? 

Reviewer #1: No

4. Have the authors made all data underlying the findings in their manuscript fully available?

Reviewer #1: Yes

5. Is the manuscript presented in an intelligible fashion and written in standard English?

Reviewer #1: Yes

6. Review Comments to the Author

Reviewer #1: All of my previous questions are well addressed. But I have new questions for the newly added section of ‘Existing genome’ measures. In Line 343~344, the statement suggests that larger E_g means a more likely "default-on" mode. This definition is not consistent with the definitions of R_a and R_m, whose larger values suggest a more likely "default-off" mode (Line 315~317 + Line 326~328). Then, with that inconsistency, in the table 1, why do they tend to show the same sign of correlation coefficients for various model parameters? I hope the authors can check the definitions of these measurements, make these measurements more consistent with each other if possible, and clarify whether these different measures lead to similar correlations in the modified manuscript.

One typo in the manuscript: L316- two equal signs in the equation?

7. PLOS authors have the option to publish the peer review history of their article (what does this mean?). If published, this will include your full peer review and any attached files.

Reviewer #1: No

---

## [Author Response · Author response to Decision Letter 1]

27 Apr 2021

Reviewer #1: All of my previous questions are well addressed. But I have new questions for the newly added section of ‘Existing genome’ measures. In Line 343~344, the statement suggests that larger E_g means a more likely "default-on" mode. This definition is not consistent with the definitions of R_a and R_m, whose larger values suggest a more likely "default-off" mode (Line 315~317 + Line 326~328). Then, with that inconsistency, in the table 1, why do they tend to show the same sign of correlation coefficients for various model parameters? I hope the authors can check the definitions of these measurements, make these measurements more consistent with each other if possible, and clarify whether these different measures lead to similar correlations in the modified manuscript.

We understand the reviewer’s confusion, and apologise for not explaining this more clearly. In all measures, a value of <1 is associated with a ‘default off’ mode. In the ‘new gene’ modes, regulatory element length is inversely related to the probability that that element is active. So length(negative elements)/length(positive) elements is <1 if the length of the negative elements is shorter than the length of the positive elements, and hence the probability of the negative elements being active is larger than the probability of the positive elements being active. For the ‘existing genome’ measures we are measuring actual activity, not the chance of activity through matching to a random sequence. So if positive/negative <1, more negative elements are active than positive ones.

This was not made clear in the text, and we have added some text to try to remedy this. We hope it is better explained now.

The wording in lines 315-317 and 326-328 was also ambiguous, and suggested that high values of Ra and Rm meant ‘default off’. In Table 1 it was correctly stated that for all measures <1 implied ‘default off’. We have corrected this wording. 

One typo in the manuscript: L316- two equal signs in the equation?

Thank you, yes this was an error, now corrected. 

We have also added a reference to the introduction to a recent Nature paper on proteins in chromatin structure in Yeast, which has some relevance.

---

## [Editor Report · Decision Letter 2]

29 Apr 2021

Evolution of default genetic control mechanisms

PONE-D-21-00488R2

Dear Dr. Bains,

We’re pleased to inform you that your manuscript has been judged scientifically suitable for publication and will be formally accepted for publication once it meets all outstanding technical requirements.

Kind regards,

Ernesto Perez-Rueda

Academic Editor

PLOS ONE

Additional Editor Comments (optional):

I consider that the manuscript is scientifically suitable for publication.
---

## [Editor Report · Acceptance letter]

4 May 2021

PONE-D-21-00488R2 

Evolution of default genetic control mechanisms 

Dear Dr. Bains:

I'm pleased to inform you that your manuscript has been deemed suitable for publication in PLOS ONE. Congratulations! Your manuscript is now with our production department. 

Kind regards, 

on behalf of

Dr. Ernesto Perez-Rueda 

Academic Editor

PLOS ONE